# Bedside Hyperspectral Imaging and Organ Dysfunction Severity in Critically Ill COVID-19 Patients—A Prospective, Monocentric Observational Study

**DOI:** 10.3390/bioengineering10101167

**Published:** 2023-10-06

**Authors:** Henning Kuhlmann, Lena Garczarek, David Künne, Kevin Pattberg, Annabell Skarabis, Mirjam Frank, Börge Schmidt, Sven Arends, Frank Herbstreit, Thorsten Brenner, Karsten Schmidt, Florian Espeter

**Affiliations:** 1Department of Anesthesiology and Intensive Care Medicine, University Hospital Essen, University Duisburg-Essen, 45147 Essen, Germany; 2Institute for Medical Informatics, Biometry and Epidemiology, University Hospital Essen, University Duisburg-Essen, 45147 Essen, Germany

**Keywords:** hyperspectral imaging, COVID-19, sepsis, ECMO, critical care

## Abstract

Hyperspectral imaging (HSI) is a non-invasive technology that provides information on biochemical tissue properties, including skin oxygenation and perfusion quality. Microcirculatory alterations are associated with organ dysfunction in septic COVID-19 patients. This prospective observational study investigated associations between skin HSI and organ dysfunction severity in critically ill COVID-19 patients. During the first seven days in the ICU, palmar HSI measurements were carried out with the TIVITA^®^ tissue system. We report data from 52 critically ill COVID-19 patients, of whom 40 required extracorporeal membrane oxygenation (ECMO). HSI parameters for superficial tissue oxygenation (StO_2_) and oxygenation and perfusion quality (NPI) were persistently decreased. Hemoglobin tissue content (THI) increased, and tissue water content (TWI) was persistently elevated. Regression analysis showed strong indications for an association of NPI and weaker indications for associations of StO_2_, THI, and TWI with sequential organ failure assessment (SOFA) scoring. StO_2_ and NPI demonstrated negative associations with vasopressor support and lactate levels as well as positive associations with arterial oxygen saturation. These results suggest that skin HSI provides clinically relevant information, opening new perspectives for microcirculatory monitoring in critical care.

## 1. Introduction

The first two years of the coronavirus disease (COVID-19) pandemic showed that SARS-CoV-2 infection and the concomitant immune dysregulation inflict a systemic microvascular disturbance leading to organ damage and failure [1]. Systemic microvascular impairment is now recognized as a hallmark feature of COVID-19-associated acute respiratory distress syndrome (ARDS) and septic shock, necessitating new microcirculation monitoring initiatives in critically ill COVID-19 patients [2]. The importance of maintaining adequate oxygen delivery and tissue perfusion, as well as preventing detrimental fluid and vasopressor effects, are cornerstones of critical care therapy. Clinicians currently lack important microcirculation feedback in hemodynamic therapy for critically ill patients, while existing methods for measuring microcirculation have a variety of limitations and unresolved complexities [3]. Consequently, bedside monitoring technologies for microcirculatory diagnostics still need to find their way into the clinical routine [4,5,6,7]. Hyperspectral imaging (HSI) is a novel non-invasive optical technology that enables bedside identification of cutaneous microcirculatory alterations and biochemical tissue analysis [8]. Cutaneous microcirculatory alterations correlate with organ dysfunction severity and respond to hemodynamic therapy in critically ill patients, making the skin an attractive organ for microcirculatory monitoring research [5,6]. HSI is increasingly studied for surgical applications [9], whereas data on HSI technologies for microcirculatory diagnostics and hemodynamic therapy in critically ill patients are limited [10,11] First experiences demonstrated the feasibility of HSI diagnostics using the TIVITA^®^ Tissue camera system (Diaspective Vision GmbH, Am Salzhaff, Germany) to detect clinically relevant microcirculatory skin alterations in critically ill patients with sepsis and major abdominal surgery [10,12]. The TIVITA^®^ Tissue camera system (Diaspective Vision GmbH, Am Salzhaff, Germany) provides non-invasive HSI for qualitative and quantitative bedside microcirculatory assessment [8] Furthermore, Dietrich et al. showed in a porcine model of hemorrhagic shock that HSI detected dynamic changes in tissue oxygenation and perfusion quality. This way, an estimation of resuscitation effectivity as well as the identification of detrimental side effects of fluid and vasopressor therapy could be provided [13]. 

The aim of this study was to evaluate possible associations between cutaneous HSI parameters and organ dysfunction severity in critically ill COVID-19 patients. This observational study was performed in a specialized ARDS ICU in a German tertiary medical center from April 2020 to April 2021. To the best of our knowledge, this is the first data on cutaneous HSI monitoring in critically ill septic COVID-19 patients with ARDS and ECMO support.

## 2. Materials and Methods

This monocentric prospective observational study was conducted in accordance with the Declaration of Helsinki at the Department of Anesthesiology and Intensive Care Medicine, University Hospital Essen, University of Duisburg-Essen, between April 2020 and April 2021. The study was approved by the Ethics Committee of the Medical Faculty of the University of Duisburg-Essen (Ethics Committee Number: 20-9242-BO) and registered in the German Clinical Trials Registry (German Clinical Trials Registry Number: DRKS-ID: DRKS00022441). The study was registered retrospectively in 2020 after the scope of the pandemic and the need for specific therapies became apparent. Data from the same collective was published by Espeter et al. [14]. The primary endpoint was the association of cutaneous HSI parameters with the severity of organ dysfunction using a modified sequential organ failure assessment (SOFA) score.

### 2.1. Hospital Setting 

The University Hospital Essen is a tertiary care medical center. The Intensive Care Unit of the Department of Anesthesiology and Intensive Care Medicine is a regional referral center for ARDS patients and is particularly specialized in extracorporeal membrane oxygenation (ECMO) therapy, with more than 100 such procedures per year. A substantial portion of patients treated in this ICU are transferred from referring hospitals by ground or air ambulance by a 24/7 on-call critical-care team equipped with a mobile ECMO unit (Cardiohelp ©, Getinge, Rastatt, Germany), enabling on-site initiation of ECMO therapy at the referring hospital if needed. Being part of the West German Center for Infectious Diseases, the ICU is specialized in treating critically ill patients with infectious diseases, including advanced isolation features like single bedrooms with negative pressure environments. During the COVID-19 pandemic, this ICU had a central role in treating critically ill COVID-19 patients within the pandemic management strategy of the University Hospital Essen [15].

### 2.2. Patient Recruitment and Study Protocol

Between April 2020 and April 2021, 54 patients were included. Patients or, in the case of unconscious patients, their legal guardians gave informed consent to participation. An independent medical consultant validated consent to study inclusion in accordance with the regulations of the medical faculty of the University Duisburg-Essen if no legal guardian was available for unconscious patients. Additionally, we obtained consent after recovery from patients initially unable to give consent. The study protocol and examination timeline are shown in Figure 1. The first HSI measurement was performed at admission to the ICU (=day 0). Further measurements were recorded on day 1, day 2, day 3, and day 7 after admission to the ICU. Four investigators were responsible for HSI measurements and the documentation of study-relevant clinical data. These results were not accessible to the treating physicians and therefore had no influence on the clinical therapy.

### 2.3. Patient Treatment 

All patients admitted to the ICU were treated according to a standardized treatment protocol published previously [15]). The ECMO indication was based on the respective recommendations of the Extracorporeal Life Support Organization (ELSO) [16]. Cannulation for veno-venous ECMO therapy was performed either bifemorally or femorojugularly. Interdisciplinary COVID-19 treatment was based on the respective recommendations for COVID-19 therapy, with these recommendations varying during the study period (April 2020–April 2021) [17].

### 2.4. HSI Measurement 

#### 2.4.1. HSI Camera System 

We used the TIVITA^®^ Tissue camera system (Diaspective Vision GmbH, Am Salzhaff, Germany) for HSI image acquisition. The camera system is a validated class I medical device and was used in accordance with the operating instructions. The system is mounted on a mobile cart with the camera attached to a continuously adjustable system, allowing for easy bedside image acquisition. To standardize image acquisition, the camera contains target optics consisting of two LED lights that overlap at 50 centimeters between the camera and the object. Illumination is provided by LED lights attached next to the camera lens. A sensor measuring ambient light informs the user about too bright or too dark ambiance. One measurement takes 6.4 s. During one measurement, four HSI parameters in varying tissue depths are obtained depending on light wavelength characteristics [8,18]:Tissue Oxygenation (StO_2_) (wavelength range: 570–590 nm, penetration depth < 1 mm).Near infrared perfusion index (NPI) (wavelength range: 655–735 nm and 825–925 nm, penetration depth: 3–5 mm).Tissue Hemoglobin index (THI) (wavelength range: 530–590 nm and 785–825 nm).Tissue Water Index (TWI) (wavelength range: 880–900 nm and 955–980 nm).

The unit for StO_2_ is percent [0–100%], while NPI, THI, and TWI are given as arbitrary units [0–100 AU]. The parameter calculation is based on the optical absorption characteristics of hemoglobin and water in specified sections of the spectrum between 500 and 1000 nm. The complex algorithms for the parameter calculation are presented in detail by Kulcke et al. [19] and Holmer et al. [8,18].

Bedside picture visualization of the investigated area is completed within 30 s. The analyzing software calculates color-coded images for each parameter to facilitate qualitative visual interpretation of parameters. Values towards the lower range (0–50) are shown as blue and green, while the higher ranges (50–100) are shown as yellow and red. The camera-specific software (TIVITA^®^TM Suite) allows for the definition of regions of interest (ROI), in which mean values for the four parameters are calculated for quantitative analysis (Figure 1).

#### 2.4.2. Image Acquisition

HSI measurements were performed without interfering with medical therapy. HSI picture acquisition conditions were standardized utilizing the camera system’s integrated stray light warning sensor. Images were acquired at the palm due to easy access in supine as well as prone-positioned ventilated ARDS patients. HSI measurements were taken from both palms, where patient positioning allowed access for picture acquisition. Hand positioning was standardized to expose an open palm. The presence of an invasive blood pressure measurement was not considered an exclusion criterion when selecting the hands for the examination. For picture analysis, we defined a region of interest (ROI) of the palm area between the wrist and proximal phalanges II-V excluding the curved surface of the thenar. To calculate the mean values in the defined ROI, we performed three HSI measurements per side. We refrained from analyzing the fingertip because of the regularly identified tissue necrosis in shock patients.

#### 2.4.3. Clinical Data Acquisition and SOFA Score Calculation

With each HSI measurement, the corresponding vital-, hemodynamic-, respiratory-, and ECMO parameters were obtained, and the SOFA score was calculated. For disease severity discrimination in ARDS patients treated with ECMO therapy, we adjusted the SOFA score and added an additional point, increasing the theoretical maximum score from 24 to 25. The SOFA score was modified to reflect sedated ICU patients as described by Vasilevskis et al. [20]. 

### 2.5. Statistical Analysis 

HSI measurements were included in the statistical analysis as follows: For each parameter, the mean of the left- and right-hand values was calculated for each study participant. If a measurement for one hand was missing, the value of the non-missing measurement was used instead. Then, the mean of the second and third measurement mean values was included in the statistical analysis to prevent bias by regression to the mean [21].

To characterize the analysis population, descriptive statistics of the variables included in the analysis were stratified by day of examination. For normally distributed continuous variables, the mean (±standard deviation [SD]) was presented, and for variables deviating from a normal distribution, the median (interquartile range [IQR]). Categorical variables were presented as numbers (%).

To visualize potential associations between HSI measures and SOFA scores across all study participants and all examination days, scatter plots with regression lines were plotted. To assess the strength of the association between HSI measures as dependent variables and SOFA score as independent variables in all study participants and across all examination days, linear mixed regression models were fitted to calculate effect size estimates (β) and 95% confidence intervals (95%-CI) with study participant as random effect additionally adjusted for sex, age, and BMI as potential confounders. The association between HSI measures and SOFA score was also stratified by examination day using linear regression models to assess trends in effect size estimates across strata. Here, r^2^ was additionally calculated to quantify the variance of HSI measures explained by the SOFA score. Additional linear mixed regression models were calculated to assess the association between HSI measures and other potential predictors such as ECMO therapy, norepinephrine dosage, and blood lactate concentration, with study participant ID as a random effect additionally adjusted for sex, age, and BMI as potential confounders. All analyses were performed using the R statistical package 4.1.1 [22].

## 3. Results

### 3.1. Characteristics of Critically Ill COVID-19 Patients

From April 2020 to April 2021, 54 critically ill COVID-19 patients were included. Two patients were subsequently excluded because they later refused to participate. According to the World Health Organization's (WHO) COVID-19 severity classification, six patients were classified as severely ill (requiring oxygen via mask or spectacles), and 46 patients with COVID-19 were classified as critically ill (requiring ventilation and/or vvECMO therapy). Forty-six patients were transferred to our department from referring hospitals due to severe ARDS [23]. In 25 cases, vvECMO therapy was initiated externally by our mobile ECMO team. Six patients were admitted directly from the emergency department or a COVID-19 intermediate care unit of Essen University Hospital due to clinical deterioration.

Patient demographic, clinical, and outcome data are presented in Appendix A. These data have already been shown by Espeter et al. in a previous study evaluating butyrylcholinesterase as a prognostic biomarker in critically ill COVID-19 patients [14]. Disease severity, resuscitation therapy, and clinical parameters from admission to our intensive care unit until day 7 after admission are shown in Table 1 for all included critically ill COVID-19 patients (*n* = 52) and in Table 2 for critically ill COVID-19 patients with vvECMO support (*n* = 40).

Missing HSI data on day 0 resulted in patients being admitted to our ICU late at night with none of the investigators familiar with the HSI camera on site to perform measurements. Missing data following day 0 results from patients not being accessible to the investigators due to medical procedures like surgery. Missing data on day 7 resulted in four patients dying and one patient ending up discharged from the ICU before day 7.

Clinical data were obtained from written charts during the COVID-19 pandemic. Missing clinical data results from parameters not being recorded primarily on charts. In some cases, parameters like peripheral oxygen saturation could not be measured due to severely impaired skin microcirculation.

### 3.2. Hyperspectral Imaging (HSI) in Critically Ill COVID-19 Patients

A representative example of HSI as color-coded images for qualitative assessment of a patient in septic shock with COVID-19 pneumonia, ARDS, and vvECMO therapy on admission to the ICU (=day 0) and on day 7 is shown in Figure 2.

The mean values of HSI parameters in critically ill COVID-19 patients from the day of admission to the ICU (day = 0) until day 7 after ICU admission are presented in Table 3. Table 3 shows HSI data for those COVID-19 patients undergoing ECMO therapy.

### 3.3. The Association between HSI Measurements and Disease Severity in Critically Ill COVID-19 Patients

Across all observation time points, we observed the strongest association with disease severity based on the SOFA score for NPI, which showed a 0.308 lower NPI score (beta: −0.308; 95% CI −0.596; −0.035) per SOFA score unit. StO_2_ showed a weaker association, with an average of 0.250 lower StO_2_ scores (beta: −0.250; −95% CI −0.630; 0.117) per SOFA score unit. THI and TWI showed a weak association, with an average THI value increase of 0.272 (beta: 0.272; 95% CI −0.106, 0.677) and an average TWI value increase of 0.205 (beta: 0.205; 95% CI −0.034, 0.457) per SOFA score unit (Table 4). Scatter plots illustrating the association of StO_2_, NPI, THI, and TWI with SOFA score, including all examination time points, are presented in Appendix A.

### 3.4. Strength of the Association between NPI and Disease Severity Increases with Length of ICU Treatment in Critically Ill COVID-19 Patients

Next, we analyzed the strength of the association between NPI and SOFA scores stratified for the respective examination time points. The strength of the association between NPI and SOFA score as well as the determination coefficient R^2^ increased during the observation time, being strongest seven days after ICU admission with 14.3% of the variance in NPI explained by the SOFA score (Figure 3).

### 3.5. Association between NPI and Disease Severity Is Stronger in Critically COVID-19 Patients without vvECMO Therapy

A difference in the strength of the association was observed between critically ill COVID-19 patients with vvECMO and without vvECMO. The association between NPI and disease severity was particularly evident in COVID-19 patients without vvECMO therapy and less strongly in patients with vvECMO (Figure 4).

### 3.6. StO_2_ and NPI Are Strongly Associated with Norepinephrine Support and Blood Lactate Levels

Additionally, we analyzed the association between HSI parameters and norepinephrine support (Table 5) as well as blood lactate concentrations (Table 5). The strongest effect estimates for associations with StO_2_ and NPI were observed for norepinephrine dose and lactate concentrations. Per µg/kg/min of norepinephrine, StO_2_ decreased by 4.621 points (95% CI −7.271; −1.963) and NPI by 3.198 points (95% CI −5.198; −1.201). An increase in blood lactate of 1 mmol/l decreased StO_2_ by 1.658 points (95% CI −2.379; −0.951) and NPI by 1.048 points (95% CI −1.619; −0.477). StO_2_ and NPI showed no association with mean arterial pressure (MAP) (Table 5). Both parameters were associated with arterial oxygen saturation (SaO_2_) (Table 5). For SaO_2_, an increase of 1% resulted in an increase of StO_2_ of 0.283 points (95% CI 0.085; 0.480) and an increase of NPI of 0.172 points (95% CI 0.011; 0.328).

## 4. Discussion

To the best of our knowledge, this is the first data on skin HSI examinations using the TIVITA^®^ Tissue camera system in critically ill patients with COVID-19 pneumonia. Furthermore, we show the first results of skin HSI monitoring in septic patients on vvECMO support. The major findings of our study are the association between HSI parameters of tissue oxygenation and perfusion quality (StO_2_ and NPI) and the severity of organ dysfunction as measured by a modified SOFA score. Tissue hemoglobin distribution (THI) and tissue water content (TWI) showed a weak relationship with the SOFA score. Additional analysis showed negative associations with vasopressor support and lactate levels as well as positive associations with arterial oxygen saturation for skin HSI oxygenation parameters.

Currently, the heterogeneity of study designs, severity of COVID-19 infection, timing of assessment during disease progression, microcirculation monitoring technologies used, and assessment of the microcirculatory region of interest leads to conflicting evidence regarding the associations between microcirculatory dysfunction and disease severity in COVID-19 [24,25]. Despite evidence of COVID-19-specific pathomechanisms causing microvascular endothelial damage and microvascular reactivity reduction along with COVID-19-associated coagulopathy [24,26] there is a clear overlap with previously described patterns of septic microcirculatory changes [24,27]. Nevertheless, an essential common feature of critically ill patients with severe COVID-19 sepsis with and without bacterial superinfection is that a heterogeneous pattern of microcirculatory dysfunction prevails and characterizes the course of the disease [1,24]. Given the incubation period of SARS-CoV-2 infection and despite the individual length of time until critical deterioration leads to admission to the ICU, the COVID-19 patients in our study showed late states of microcirculatory alterations. Most patients in this study were in an advanced phase of their sepsis course. They were included after near-fatal acute respiratory failure and severe shock and required resuscitation therapy and empiric antibiotic treatment for secondary bacterial infections. We, therefore, propose that our data do not reflect a COVID-19-specific HSI pattern but rather a mixed picture of severe COVID-19 sepsis together with bacterial sepsis.

At this point, it is important to mention that HSI is not able to quantify perfusion directly, i.e., to measure flow in vessels. Its main functionality is to show the effect of perfusion and oxygenation on tissues [8,19]. We observed persistently low and even decreasing StO_2_ and NPI values and an increasing trend in THI values during the observation period. TWI values were in an elevated range, suggesting a persistent increase in tissue water. The HSI measurements suggest that respiratory as well as vvECMO therapy with the restoration of sufficient arterial oxygenation and the achievement of macrocirculatory stabilization did not restore adequate skin oxygenation and perfusion quality. Our findings in ECMO patients compare well with the results of Carsetti et al. demonstrating sublingual microcirculatory alterations in COVID-19 ARDS patients with vvECMO support. Based on their observation, they conclude that vvECMO by itself might not improve microvascular alterations and even propose that ECMO might have a detrimental effect on microcirculation due to an increase in inflammation because of the prolonged contact of blood with extracorporeal circulation [28]. The increased tissue water accumulation indicated by TWI in our patients might indirectly point to tissue edema formation due to sepsis-associated increased microvascular permeability. The formation of tissue edema certainly aggravates oxygen deficits in tissues. Therefore, in our patients, the joint occurrence of increased TWI values with reduced StO_2_ and NPI values is to be expected or to be pathophysiologically justified. Our results of HSI parameter patterns and the associated mean values of StO_2_, NPI, THI, and TWI are comparable to the results of Dietrich et al. [10]. They proposed that septic patients present a specific skin HSI pattern indicative of decreased microcirculatory oxygenation and perfusion quality in combination with microcirculatory blood pooling and increased tissue water content. Non-surviving septic patients showed low StO_2_ and NPI values despite prior hemodynamic stabilization. Accordingly, Dietrich et al. suggested this to indicate deficits in skin oxygenation and perfusion quality despite the achievement of hemodynamic stability [10]. In addition, Dietrich et al. observed a sustained increase in TWI with a positive correlation 72 h after ICU admission with the SOFA score [10]. Additionally, the THI allowed a prognostic prediction of 28-day mortality in patients with sepsis. However, when interpreting our data, it is important to note that patients in our study exhibited microcirculatory alterations due to severe ARDS and profound or prolonged septic shock in a late stage of COVID-19-associated sepsis. This limits the comparability to previous studies using HSI for skin evaluation in early sepsis phases [10,29]. Furthermore, the large number of vvECMO patients in our collective does not allow a transfer of the findings to sepsis patients without vvECMO without caution. In our critically ill COVID-19 patients, the NPI showed a strong inverse relationship with the SOFA score. Interestingly, we observed an increase in the explanatory power of this relationship between NPI and SOFA over the observation period. We propose that NPI could alert clinicians to latent tissue oxygen deprivation and provide information about the overall effectiveness of resuscitation during treatment. On the other hand, NPI could aid in bedside risk stratification by providing a general assessment of the severity of organ dysfunction and response to therapy. The observed strong negative association of StO_2_ and NPI with lactate level and the positive association with arterial oxygen saturation support our hypothesis that skin NPI might point to systemic oxygenation deficits. Therefore, NPI could be indicative of a potentially irreversible microcirculatory dysfunction and could represent an integral time- and resuscitation therapy-dependent component of microcirculatory feedback in a multimodal monitoring strategy for critically ill patients.

An unexpected result of the subgroup analysis is that the association between NPI and SOFA score was weaker in patients with vvECMO support than in patients without. A possible explanation for our result is that the modified SOFA score and NPI demonstrate opposite effects, confounding the association between disease severity and NPI in vvECMO patients. To represent the need for vvECMO therapy as an expression of the most severe form of respiratory failure and not underestimate the degree of respiratory failure due to paO_2_/FiO_2_, we modified the SOFA score and accordingly assigned an additional point in the category: lungs on vvECMO therapy (5 = points). This modification results in patients with vvECMO receiving one more SOFA point and thus being represented as sicker. The NPI, on the other hand, should be positively influenced by vvECMO therapy, which restores arterial oxygenation. We propose that this limitation underlies the weaker association of NPI with SOFA score in patients with vvECMO therapy compared to patients without vvECMO support. However, this finding requires further analysis in follow-up studies, including a detailed examination of advanced hemodynamic monitoring with cardiac index assessment, HSI before and after ECMO initiation, and shunt fraction under ECMO assistance.

Considering the wavelength of NPI (655–735 nm and 825–925 nm), our results are consistent with previous HSI results in septic patients by Kazune et al., who reported an association between lower microcirculatory skin oxygenation (wavelength range 450–820 nm) and higher SOFA values [11]. The results of lower HSI-measured skin oxygenation in septic patients by Dietrich et al. are also corroborated by Kazune et al., who showed a significantly lower HSI-measured skin microcirculatory oxygenation in septic patients with a mottling score of 2 compared to those with a mottling score of 0 [29]. Kazune et al. showed that cutaneous HSI allows earlier and more specific identification of patients with emergent heterogeneity based on the distribution pattern of hemoglobin concentration and microcirculatory oxygenation of the skin compared with the mottling score [29]. This mirrors the interpretation of changes in THI values and their spatial distribution by Dietrich et al., who proposed that THI patterns might be indicative of disturbed or stagnant perfusion in the skin microcirculation. Comparable to the HSI results of Kazune et al., we observed recurrent HSI images that showed heterogeneity of perfusion and oxygenation in the skin areas examined. We assume that bedside HSI measurements could overcome the unresolved drawbacks of semiquantitative, examiner-dependent clinical assessments such as capillary refill time or mottling score for microcirculation diagnostics in septic patients.

Dietrich et al. and Kazune et al. previously proposed that skin HSI could provide microcirculatory feedback to macrocirculatory targeted interventions [11,13]. Kazune et al. showed in septic patients that raising the mean arterial blood pressure by 20 ± 5 mmHg from baseline (median baseline MAP above 65 mmHg) by increasing norepinephrine dosage improved the HSI-measured microcirculatory oxygenation of the skin. The authors propose that HSI technology detects individual changes in skin oxygenation in response to increased perfusion pressure. More improvement in oxygen saturation values was observed in patients with higher SOFA scores, whereas lower skin oxygenation at MAP of 85 mm Hg was associated with higher mortality [11]. Kazune et al. argued that monitoring skin microcirculation is especially important if higher vasopressor doses are used to achieve higher MAP target values. We did not examine HSI changes following a hemodynamic intervention protocol like Kazune et al. [11]. All patients in our study had mean arterial pressures > 65 mmHg with varying norepinephrine support at the time of skin HSI assessment. Our analysis showed no association between mean arterial pressure and StO_2_ or NPI. Still, it is intriguing that given the 28-day mortality in our patients, we observed a similar occurrence of mean MAP > 85 mmHg and decreased HSI skin oxygenation. StO_2_ as well as NPI demonstrated a strong negative association with noradrenaline dosage in our patients. We showed that HSI-measured tissue oxygenation and perfusion quality decreased with both increasing norepinephrine dosage and blood lactate concentration. We propose that future integration of skin HSI data into a multimodal assessment could warn clinicians of adverse therapy effects and allow for balancing hemodynamic goals and therapy. Our proposal reflects experimental results from Dietrich et al., who previously demonstrated in a porcine hemorrhagic shock model that HSI detected dynamic changes in tissue oxygenation and perfusion quality during shock and could indicate the effectiveness of resuscitation. Interestingly, they reported a correlation between skin and kidney HSI parameters, suggesting that skin HSI could provide inferences about microcirculation in different organ systems. They proposed a differentiated analysis of StO_2_ and NPI, THI, and TWI that could enable tissue perfusion-guided therapy and identify adverse effects of vasopressor and fluid therapy, respectively.

To bridge the gap from research to clinical application for microcirculatory monitoring, the application of artificial intelligence is an emerging field of interest. Previously, Hilty et al. showed that a deep learning-based model successfully differentiates critically ill COVID-19 patients from healthy volunteers in sublingual microcirculation microscopy [30]. Due to its technical features to analyze tissue morphology and composition, HSI is increasingly evaluated for automated artificial intelligence applications [9,31]. A pioneering next step in skin HSI analysis will be to combine automated HSI image acquisition and raw spectral data analysis with machine learning algorithms. Studier-Fischer et al. demonstrated the feasibility of HSI for machine learning analysis by developing a tissue atlas of twenty different porcine organs and tissue types and defining HSI-based spectral organ fingerprinting [32]. Therefore, we hypothesize that artificial intelligence-assisted skin HSI raw data analysis could provide user-friendly bedside diagnostic support and therapeutic guidance in critical care. Furthermore, incorporating additional statistical analyses, such as techniques for identifying outliers and conducting regression analysis in the presence of variations, could enhance the precision of data interpretation.

Several limitations should be acknowledged for the interpretation of our results: Due to the novelty of HSI technology, there is no standardization in measured spectral ranges, picture acquisition technology, or analyzing software algorithms. Furthermore, there are no standardized definitions of normal ranges of skin HSI parameters for different patient groups, which limits comparisons between HSI data [10,11,29]. Against this background, we consider our analysis strategy, which is adapted to possible confounding factors (gender, age, and BMI), to be a strength of our study. Furthermore, there is no consensus as to which anatomical site is most appropriate or meaningful for HSI skin examinations. Dietrich et al. evaluated the palms, fingertips, and suprapatellar knee area as HSI examination sites [10]. Kazune et al. performed HSI examinations in an area above the patella to allow correlation of their measurements with clinical skin patches [29]. The ARDS patients in our group had to be repeatedly placed in the prone position and often had necrosis of the fingers after a prolonged stay in the intensive care unit, so that only the palm was reliably accessible. Although no patient with dark skin color was included in our study, a possible influence of dark skin on the results of HSI measurement can be considered a limitation. Furthermore, the individual skin condition (e.g., epidermis thickness) could influence the results of HSI. A potential limitation of performing palm HSI measurements in the intensive care setting is the prevalence of radial arterial catheters, which might result in iatrogenic hypoperfusion in the dependent palm area. To minimize the potential influence of arterial catheters, we opted to obtain HSI measurements from both hands. For the HSI picture analysis, we excluded the curved surface of the thenar to reduce light scattering and reflection artifacts on HSI parameters. We used the TIVITA^®^ Tissue software analysis tool to freely mark the region of interest according to anatomical landmarks. A strength of our HSI examination strategy is that we always performed three consecutive HSI measurements and included only the second and third measurements to prevent statistical bias by regression to the mean.

This study was conducted in the context of the COVID-19 pandemic, leading to a selection bias of severely critical-ill COVID-19 patients. We observed a 28-day mortality rate of 64% in the patient cohort of our study. Comparable mortality rates of up to 71% in critically ill COVID-19 patients with vvECMO therapy were reported in a large, nation-wide German study with over 10,000 COVID-19 patients [33]. The analyses of an observational study at 26 German ECMO centers show that in high-volume centers such as ours (defined as specialized ECMO centers with more than 50 vvECMO therapies in 2019), 38% of COVID-19 vvECMO patients survived [34]. The primary endpoint of our analysis was the association of skin HSI parameters with disease severity in the first seven days following admission to our ICU. Due to clinical confounding factors such as initiation of palliative treatment in therapy-refractory lung failure and termination of ECMO therapy, we refrained from a detailed analysis of the HSI parameters with 28-day mortality in our collective. Our cohort of critically ill COVID-19 patients is heterogeneous regarding respective recruitment time points in their individual illness time course. We were not able to obtain detailed pre-treatment data for most of our patients during the COVID-19 pandemic. Ultimately, however, we cannot exclude the possibility that there are differences between patients with sepsis from COVID-19 or between patients with COVID-19 and bacterial sepsis. Vascular occlusion testing followed by reactive hyperemia is recommended to determine vascular responsiveness. Given the major medical and logistical challenges of the COVID-19 pandemic, we were unable to integrate such a test for vascular reactivity. In addition, data on extended hemodynamic monitoring (e.g., cardiac index) or the administration of blood transfusions during the observation period were not documented in direct temporal relation to the HSI measurements. In follow-up studies, further questions on such factors potentially influencing the microcirculation must be investigated.

## 5. Conclusions

Going forward, the challenge is to provide noninvasive, simple, reliable, and bedside assessment as well as quantitative analysis of the microcirculation of critically ill patients. We demonstrate an association between HSI parameters of tissue oxygenation and perfusion quality (StO_2_ and NPI) in critically ill COVID-19 patients and the severity of organ dysfunction. Consistent with previous studies, our results suggest that HSI may represent a missing piece in the puzzle for bedside microcirculatory monitoring. Differentiated skin HSI patterns might facilitate immediate feedback about the status of the microcirculation in critically ill patients, providing important diagnostic information to clinicians. In conclusion, HSI may enrich future bedside monitoring of microcirculation in critically ill patients, thereby enabling individualized management of hemodynamic therapy.

## Figures and Tables

**Figure 1 bioengineering-10-01167-f001:**
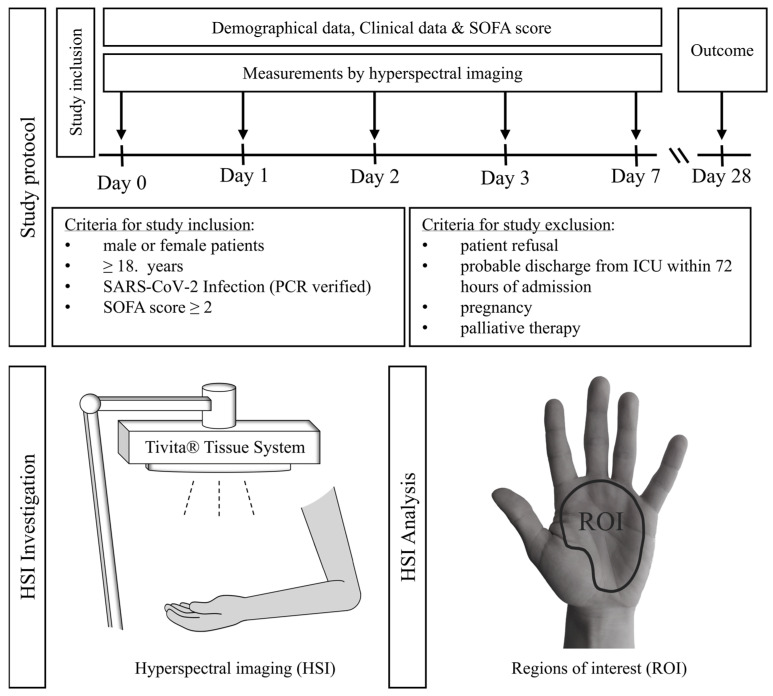
Study protocol: Following patient recruitment, palmar hyperspectral imaging (HSI) measurements were performed from day 0 (ICU admission), days 1–3, and day 7 once daily. With each HSI measurement, the corresponding vital-, hemodynamic-, respiratory-, and ECMO parameters were obtained, and the SOFA score was calculated. Mortality was assessed on day 28 following ICU admission. For HSI measurements, the TIVITA^®^ Tissue camera system (Diaspective Vision GmbH, Am Salzhaff, Germany) was used. For picture analysis, we defined a region of interest (ROI) of the palm area between the wrist and proximal phalanges II-V, excluding the curved surface of the thenar.

**Figure 2 bioengineering-10-01167-f002:**
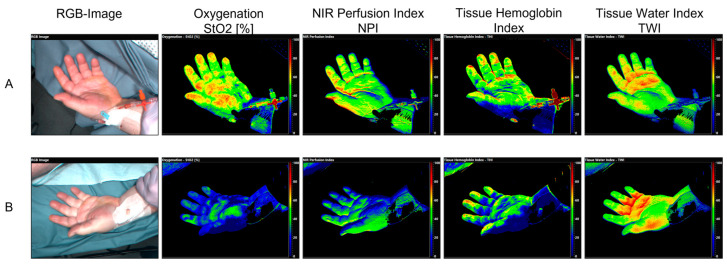
Hyperspectral imaging (HSI) of a patient in septic shock with COVID-19 pneumonia, ARDS, and vvECMO therapy (**A**) Within 24 h after admission to the intensive care unit (=day 0) and (**B**) 12 h before death on the seventh day in the ICU (day 7). Changes in tissue oxygenation (StO_2_ [%]), near-infrared perfusion index (NPI [AU]), tissue hemoglobin index (THI [AU]), and tissue water index (TWI [AU]) are indicated by the corresponding color coding (red/yellow areas indicate high values, green/blue areas indicate lower values). A heterogeneous distribution pattern of individual HSI parameters on admission and on day 7 after ICU admission is revealed by color-coded imaging. Tissue oxygenation (StO_2_) and near-infrared perfusion index (NPI) are high on admission (upper row; corresponding to red/yellow color coding) and decrease markedly until day 7 (lower row; green/blue color coding); tissue hemoglobin index (THI) also decreases from admission until day 7. In contrast, the tissue water index (TWI) increased during the observation period.

**Figure 3 bioengineering-10-01167-f003:**
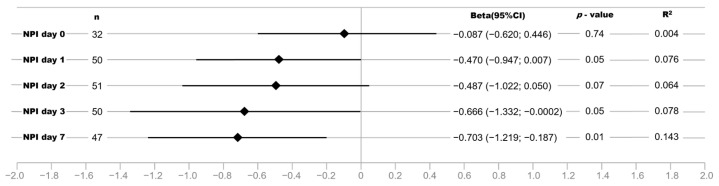
Beta estimates and 95% confidence intervals (95% CI) for the association of NPI with SOFA score in linear regression models (adjusted for age, sex, and BMI) stratified by examination at each time point (day 0, day 1, day 2, day 3, and day 7).

**Figure 4 bioengineering-10-01167-f004:**
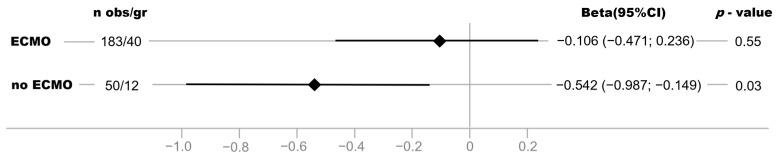
Beta estimates and 95% confidence intervals (95% CI) for the association of NPI with SOFA score in linear regression models (adjusted for age, sex, and BMI) stratified by ECMO status, including all examination time points (day 0 to day 7).

**Table 1 bioengineering-10-01167-t001:** Disease severity, resuscitation therapy, and clinical parameters of all critically ill COVID-19 patients (*n* = 52) Values are expressed as # median (interquartile range) or as * mean (±standard deviation). The number in brackets [ ] shows the number of patients with missing data.

Disease Severity, Resuscitation Therapy and Clinical Parameters of All Critically Ill COVID-19 Patients (*n* = 52)					
	Day 0	Day 1	Day 2	Day 3	Day 7
Disease severity					
SOFA Score *	12 (± 5) [11]	12 (± 4) [0]	12 (± 4) [0]	12 (± 4) [0]	13 (± 5) [4]
**Resuscitation Therapy**					
Fluids (ml per 24 h) *	234 (± 1505) [12]	−816 (± 1628) [0]	−482 (± 2217) [0]	−587 (± 1277) [1]	153 (± 2055) [5]
Norepinephrine dose (mcg/kg/min) #	0.16 (0.08–0.40) [20]	0.10 (0.04–0.19) [16]	0.04 (0.02–0.10) [15]	0.06 (0.02–0.10) [20]	0.06 (0.02–0.20) [12]
**Clinical Parameters**					
Lactate (mmol/L) #	1.7 (1.4–2.3) [11]	1.6 (1.2–1.9) [0]	1.4 (1.0–1.9) [0]	1.2 (0.9–1.8) [0]	1.5 (0.9–1.9) [4]
Hemoglobin (g/dl) #	11.2 (9.3–13.0) [11]	9.8 (8.9–12.4) [0]	9.0 (8.1–11.4) [0]	9.1 (8.5–10.6) [0]	9.2 (8.4–10.0) [4]
Temperature (Degree Celsius (°C)) #	37.0 (36.5–37.4) [11]	37.0 (36.6–37.4) [0]	37.0 (36.6–37.2) [0]	37.1 (36.8–37.4) [0]	37.0 (36.8–37.4) [4]
Mean arterial pressure (mmHg) *	90 (± 13) [11]	88 (± 14) [0]	88 (± 13) [0]	88 (± 13) [0]	80 (± 12) [4]
Heart rate (beats/min) *	92 (± 22) [11]	82 (± 16) [0]	81 (± 20) [0]	79 (± 14) [0]	85 (± 15) [4]
pH value #	7.37 (7.28–7.42) [11]	7.40 (7.35–7.46) [0]	7.42 (7.36–7.46) [0]	7.43 (7.39–7.45) [0]	7.39 (7.35–7.44) [3]
Partial pressure of carbon dioxide (mmHg) #	49 (42–61) [11]	49 (43–54) [1]	48 (42–54) [0]	47 (41–54) [0]	47 (42–53) [3]
Partial pressure of oxygen (mmHg) #	82 (69–101) [11]	78 (68–88) [0]	75 (64–85) [0]	73 (66–83) [0]	74 (67–86) [4]
paO_2_/FiO_2_ #	140 (83–210) [11]	192 (136–243) [0]	184 (122–220) [0]	179 (120–245) [0]	156 (115–220) [4]
Peripheral oxygen saturation (%) #	96 (94–99) [11]	95 (94–97) [0]	96 (93–97) [0]	95 (93–96) [0]	94 (93–97) [6]

**Table 2 bioengineering-10-01167-t002:** Disease severity, resuscitation therapy, and clinical parameters of all critically ill COVID-19 patients with vvECMO (*n* = 40) Values are expressed as # median (interquartile range) or as * mean (± standard deviation). The number in brackets [ ] shows the number of patients with missing data.

Disease Severity, Resuscitation Therapy and Clinical Parameters of Critically Ill COVID-19 Patients with vvECMO (*n* = 40)					
	Day 0	Day 1	Day 2	Day 3	Day 7
Disease severity					
SOFA Score *	14 (± 3) [9]	13 (± 2) [0]	13 (± 2) [0]	13 (± 2) [0]	14 (± 4) [2]
**Resuscitation Therapy**					
Fluids (ml per 24 h) *	375 (± 1478) [10]	−924 (± 1675) [0]	−566 (± 2435) [0]	−531 (± 1310) [0]	317 (± 2186) [3]
Norepinephrine dose (mcg/kg/min) #	0.17 (0.09–0.40) [13]	0.10 (0.04–0.20) [10]	0.04 (0.02–0.10) [10]	0.06 (0.02–0.11) [15]	0.07 (0.03–0.20) [8]
ECMO blood flow (L/min) #	4.2 (3.8–4.6) [12]	3.9 (3.3–4.7) [5]	4.2 (3.1–4.8) [5]	3.8 (3.2–4.8) [5]	4.1 (3.1–4.9) [7]
ECMO air flow (L/min) #	3 (2.00–6) [12]	4 (3–6) [5]	4 (3–6) [5]	4 (2–6) [9]	4 (2–6) [9]
**Clinical Parameters**					
Lactate (mmol/L) #	1.9 (1.5–2.4) [9]	1.6 (1.2–1.9) [0]	1.4 (1.0–1.9) [0]	1.2 (0.9–1.7) [0]	1.5 (0.9–1.8) [2]
Hemoglobin (g/dl) #	10.3 (9.3–12.3) [9]	9.6 (8.9–11.1) [0]	8.9 (8.1–9.8) [0]	9.0 (8.3–9.7) [0]	9.1 (8.3–9.5) [2]
Temperature (Degree Celsius (°C)) #	37.0 (36.4–37.4) [9]	37.0 (36.7–37.2) [0]	37.0 (36.4–37.2) [0]	37.0 (36.8–37.4) [0]	37.1 (37.0–37.4) [2]
Mean arterial pressure (mmHg) *	90 (± 12) [9]	89 (± 14) [0]	88 (± 13) [0]	88 (± 13) [0]	81 (± 14) [2]
Heart rate (beats/min) *	93 (± 22) [9]	83 (± 16) [0]	84 (± 21) [0]	80 (± 15) [0]	86 (± 16) [2]
pH value #	7.34 (7.24–7.40) [9]	7.36 (7.36–7.45) [0]	7.42 (7.36–7.45) [0]	7.42 (7.39–7.45) [0]	7.38 (7.35–7.45) [2]
Partial pressure of carbon dioxide (mmHg) #	52 (44–62) [9]	49 (44–56) [1]	48 (43–53) [0]	49 (43–54) [0]	47.8 (42–52) [2]
Partial pressure of oxygen (mmHg) #	82 (68–101) [9]	79 (54–90) [0]	76 (69–89) [0]	74 (69–89) [0]	73 (65–88) [2]
paO_2/_FiO_2_ #	107 (79–215) [9]	192 (126–243) [0]	184 (122–220) [0]	179 (120–245) [0]	156 (120–221) [2]
Peripheral oxygen saturation (%) #	95 (92–98) [9]	96 (94–97) [0]	95 (93–97) [0]	95 (93–97) [0]	94 (91–96) [2]

**Table 3 bioengineering-10-01167-t003:** HSI parameters in all critically ill COVID-19 patients (*n* = 52) and those patients with ECMO therapy (*n* = 40). Values are expressed as the mean (± standard deviation). The number in brackets [ ] shows missing data. StO_2_- Tissue oxygenation, THI- Tissue hemoglobin index, NPI- Near infrared perfusion index, TWI- Tissue water index.

**HSI Parameters in All Critically Ill COVID-19 Patients (*n* = 52).**					
	**Day 0**	**Day 1**	**Day 2**	**Day 3**	**Day 7**
StO_2_	57.05 (± 12.02) [20]	55.23 (± 11.47) [2]	57.62 (± 10.23) [1]	55.86 (± 10.57) [2]	51.82 (± 11.38) [5]
THI	30.55 (± 10.83) [20]	29.83 (± 10.64) [2]	31.38 (± 11.92) [1]	34.92 (± 10.33) [2]	37.10 (± 11.20) [5]
NPI	48.06 (± 7.26) [20]	48.24 (± 6.99) [2]	49.41 (± 7.82) [1]	47.28 (± 8.56) [2]	45.30 (± 8.40) [5]
TWI	58.10 (± 7.09) [20]	58.72 (± 6.81) [2]	57.56 (± 7.28) [1]	57.30 (± 7.54) [2]	57.40 (± 6.49) [5]
**HSI Parameters in Critically Ill COVID-19 Patients with ECMO (*n* = 40)**					
	**Day 0**	**Day 1**	**Day 2**	**Day 3**	**Day 7**
StO_2_	55.80 (± 12.91) [15]	54.08 (± 12.19) [1]	57.05 (± 11.08) [1]	55.40 (± 11.02) [0]	50.99 (± 12.17) [1]
THI	30.37 (± 12.04) [15]	32.26 (± 12.79) [1]	31.38 (± 11.92) [1]	34.98 (± 10.98) [0]	38.24 (± 10.42) [1]
NPI	48.12 (± 7.81) [15]	48.40 (± 7.86) [1]	49.41 (± 7.82) [1]	47.04 (± 9.07) [0]	44.54 (± 7.36) [1]
TWI	58.94 (± 6.77) [15]	58.37 (± 6.80) [1]	57.56 (± 7.28) [1]	58.33 (± 7.28) [0]	58.87 (± 5.90) [1]

**Table 4 bioengineering-10-01167-t004:** Results of linear mixed model regression analysis for the association of StO_2_, NPI, THI, and TWI with SOFA score in separate regression models including all examination time points (day 0, day 1, day 2, day 3, and day 7) with study patient ID as random effects additionally adjusted for sex, age, and BMI.

	*n* obs/gr	beta	95% CI	*p* Value
StO_2_	233/52	−0.250	(−0.630; 0.117)	0.19
NPI	233/52	−0.308	(−0.596; −0.035)	0.03
THI	233/52	0.272	(−0.106; 0.677)	0.17
TWI	233/52	0.205	(−0.034; 0.457)	0.10

**Table 5 bioengineering-10-01167-t005:** Beta estimates and 95% confidence intervals (95% CI) for the associations of norepinephrine dosage (µg/kg/min), blood lactate concentration, mean arterial pressure, and arterial oxygen saturation with StO_2_, NPI, THI, and TWI, including all examination time points (day 0 to day 7), using separate regression models adjusted for sex, age, and BMI.

**Association of StO_2_, NPI, THI, TWI with Norepinephrine Dosage**				
	** *n* ** **obs/gr**	**beta**	**95% CI**	***p*** **Value**
StO_2_	167/49	−4.621	(−7.271; −1.963)	0.001
NPI	167/49	−3.198	(−5.191; −1.201)	0.002
THI	167/49	−1.630	(−4.483; 1.322)	0.27
TWI	167/49	0.683	(−1.228; 2.595)	0.49
**Association of StO_2_, NPI, THI, TWI with Lactate**				
	** *n* ** **obs/gr**	**beta**	**95% CI**	***p*** **Value**
StO_2_	230/52	−1.658	(−2.370; −0.951)	0.0000
NPI	230/52	−1.048	(−1.619; −0.477)	0.0004
THI	230/52	−0.722	(−1.511; 0.082)	0.08
TWI	230/52	−0.038	(−0.539; 0.464)	0.88
**Association of StO_2_, NPI, THI, TWI with MAP**				
	** *n* ** **obs/gr**	**beta**	**95% CI**	***p*** **Value**
StO_2_	229/52	0.060	(−0.025; 0.145)	0.17
NPI	229/52	0.031	(−0.037; −0.097)	0.37
THI	229/52	−0.060	(−0.152; 0.031)	0.20
TWI	229/52	0.067	(0.009; 0.123)	0.02
**Association of StO_2_, NPI, THI, TWI with SaO_2_**				
	** *n* ** **obs/gr**	**beta**	**95% CI**	***p*** **Value**
StO_2_	229/52	0.283	(0.085; 0.480)	0.005
NPI	229/52	0.172	(0.011; 0.328)	0.03
THI	229/52	−0.168	(−0.384; 0.050)	0.13
TWI	229/52	−0.004	(−0.141; 0.131)	0.96

## Data Availability

The datasets generated during and/or analyzed during the current study are available from the corresponding author on reasonable request.

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
