# Peer review of "Bedside Hyperspectral Imaging and Organ Dysfunction Severity in Critically Ill COVID-19 Patients—A Prospective, Monocentric Observational Study"

_bioengineering, 2023, doi:10.3390/bioengineering10101167_

Round 1
Reviewer 1 Report
The manuscript reports a longitudinal study on microcirculation performed on critically ill COVID-19 patients in intensive care unit using a commercial hyperspectral camera (TIVITA, class I medical device) and algorithms provided by the manufacturer in the scientific literature.
The purpose of the work is to demonstrate associations between skin microcirculation parameters and severity of organ dysfunction to support clinicians in the management of hemodynamic therapy.
Four parameters were mapped and averaged on a ROI of the hand palms of 52/40 patients and statistical analyses were applied to demonstrate the association between the peripheral hemodynamic parameters and a clinical score for organ failure.
The manuscript appears to me appropriate in terms of methods and scope of the study. However, being a physicist and an engineer, notwithstanding my experience in biomedical imaging and biophotonics, I cannot estimate its clinical relevance.
My first observation concerns the appropriateness of the manuscript for the Bioengineering journal. In fact, I did not see in the manuscript either a technical development or an original method for data analysis beyond what is already reported in the scientific literature. Therefore, I think the manuscript is more suitable for a medical journal.
That said, I note that hemodynamic parameters were assessed using algorithms with calibration factors validated by the TIVITA team on healthy volunteers using a device designed for muscle oxygen monitoring (MOXY, Fortiori Design), which is not a medical device. Mapping “absolute” hemodynamic parameters, mainly in the tissue bulk, requires time-resolved methods to disentangle scattering from absorption. Even though only superficial or sub-surface parameters were determined, I wonder whether the algorithms tuned on health people are still valid when measurements are taken on seriously ill patients, whose response may be different from that used for validation.
At any rate, the meaning of the parameters measured in this study should be clearly described for the benefit of the bioengineer community who may not be familiar with these parameters. In addition, a justification of the algorithms used to calculate them must be provided. In my opining, a few reference to the scientific literature in not enough for the readers of the journal.
The near infrared perfusion index (NPI) appears to be the most important parameter in this study. I therefore express some concerns on the reliability of the associations between HSI findings and SOFA because the Authors reports that “HSI does not measure perfusion directly but allows spatial display of oxygenation and perfusion quality at different depths of the tissue layer”. This statement confuses me because I don’t understand how a CW camera can differentiate any parameter in depth. In my opinion the authors should comment on this.
I’m not in the condition of commenting the clinical relevance of the “associations” (why not correlations?) mentioned in this study. However, I note that the statistical significance of the association between NPI and SOFA is very weak for patients with vvECMO (Figure 4), who account for the majority of subjects. The Authors comment that “the large number of vvECMO patients in our collective does not allow a transfer of the findings to sepsis patients without vvECMO without caution”. As I said, I am not a physician, however the strong difference in the outcome between patients with and without vvECMO rises some doubt on the statistical analyses performed on the whole dataset which is inhomogeneous, although the Authors modified the SOFA for patients with vvECMO. I think that this issue deserves additional explanation to support the overall relevance of the study.
In conclusion, I believe that HSI could be an important instruments for surgery support and therapy monitor. In this sense the manuscript is worth publishing.
Minor comment:
In supplementary Figure 1 letters (A, B, ...) are missing and slopes do not seem to correspond to numbers in Table 4.
Reviewer 2 Report
This is an interesting and new study on the application of an actively developing hyperspectral imaging method. Undoubtedly, the degree of novelty and relevance of the work for the field in question and the broader scientific community is sufficient to justify the publication in Bioengineering. A detailed description of the medical aspects and discussion leaves a pleasant impression.
However, there are several points which should be addressed before acceptance of the manuscript.
1. Have similar studies been conducted using other methods of microcirculation monitoring, e.g., laser Doppler flowmetry, laser speckle contrast imaging, capillaroscopy, etc.? Please consider this.
2. Why was the palm area chosen? With such a measurement area, individual variability in the thickness of the stratum corneum and epidermis could have a strong influence.
3. Please provide more detailed technical information: the wavelength range of the sources, the algorithms for calculating the declared parameters, the accuracy of their measurement.
4. Please add Refs confirming the specified penetration depths.
5. Consider transferring Tables 1 and 2 to Supplementary, and Figure S1 to the main file.
6. In Tables 3a and 3b, the values are given with two decimal places. This is a very high accuracy for such measurements. Please round the numbers.
Reviewer 3 Report
The topic of the authors' research is interesting and actual. The manuscript is well structured. However, some remarks should be corrected before the manuscript is accepted.
I carefully read this manuscript. The authors have presented the results of their research correctly. The methods are described clearly. Usually, I write many remarks. However, in this instance, I have no principal questions for the authors. They did not critically analyse the research in this subject area. I have recommended the authors to add additional section. In general, the content of the manuscript is qualitative, and, in my mind, after a minor correction, the manuscript can be accepted for publication.
1. I think that this direction is interesting for many research groups. For this reason, it is necessary to add the section "Literature Survey", there to reflect the current research in this subject area and to allocate the unsolved parts of the general problem.
2. Table 1, Table 2 and Tables 5. Please, correct the size of the text for better readability.
Reviewer 4 Report
The manuscript presents the statistical analysis of the cohort of COVID-19 patients that were subjected to examination of palms with hyperspectral camera that measures several parameters. The results of regression models estimation were interpreted as association of SOFA score with different metrics.
The paper is well written and easy to follow. The results are clear and the conclusions are supported by the presented analysis. The minor revision is needed to convert the included tables from image to text form.
As correctly stated by the authors the presented analysis is to be continued and different methods including machine learning can be applied. Looking at the variability of data in Fig.S1 it may also be useful to try some outlier detection methods and regression analysis in differences.
Round 2
Reviewer 1 Report
I acknowledge that the authors have responded to my criticisms. Some revisions also improved the manuscript, which, in my opinion, can be published.
Reviewer 2 Report
The authors responded to all the comments. I recommend to accept this manuscript.
Reviewer 3 Report
Thanks, I have no other questions
Reviewer 4 Report
The authors carefully addressed the suggestions